# Benzenesulfonamides Incorporating Hydantoin Moieties Effectively Inhibit Eukaryoticand Human Carbonic Anhydrases

**DOI:** 10.3390/ijms232214115

**Published:** 2022-11-15

**Authors:** Morteza Abdoli, Viviana De Luca, Clemente Capasso, Claudiu T. Supuran, Raivis Žalubovskis

**Affiliations:** 1Institute of Technology of Organic Chemistry, Faculty of Materials Science and Applied Chemistry, Riga Technical University, P. Valdenaiela 3, LV-1048 Riga, Latvia; 2Department of Biology, Agriculture and Food Sciences, Institute of Biosciences and Bioresources, Via Pietro Castellino 111, 80131 Napoli, Italy; 3NEUROFARBA Department, Pharmaceutical and Nutraceutical Section, University of Florence, Via Ugo Schiff 6, 50019 Florence, Italy; 4Latvian Institute of Organic Synthesis, Aizkraukles 21, LV-1006 Riga, Latvia

**Keywords:** carbonic anhydrase inhibitors, sulfonamides, hydantoin

## Abstract

A series of novel 1-(4-benzenesulfonamide)-3-alkyl/benzyl-hydantoin derivatives were synthesized and evaluated for the inhibition of eukaryotic and human carbonic anhydrases (CAs, EC 4.2.1.1). The prepared compounds were screened for their hCA inhibitory activities against three cytosolic isoforms as well as two β-CAs from fungal pathogens. The best inhibition was observed against hCA II and VII as well as *Candida glabrata* enzyme CgNce103. hCA I and *Malassezia globosa* MgCA enzymes were, on the other hand, less effectively inhibited by these compounds. The inhibitory potency of these compounds against CAs was found to be dependent on the electronic and steric effects of substituent groups on the N3-position of the hydantoin ring, which included alkyl, alkenyl and substituted benzyl moieties. The interesting results against CgNce103 make the compounds of interest for investigations in vivo as potential antifungals.

## 1. Introduction

Due to the involvement of enzymes in many pathological conditions, their inhibitors are recognized as promising targets for developing novel drugs [1,2]. Interestingly, greater than one-third of current drug discovery pipelines are focused on enzyme drug targets and half of all marketed drugs are enzyme inhibitors [3]. Carbonic anhydrases (CAs, E.C.4.2.1.1) are an important family of metalloenzymes that assist the reversible interconversion of carbon dioxide and water to bicarbonate and proton (CO_2_ + H_2_O ⇋ HCO_3_^−^ + H^+^) and thereby play fundamental roles in many processes such as respiration, electrolyte secretion, pH homeostasis, and bone resorption [4,5,6]. They are, therefore, a common and valuable drug target for the treatment or prevention of a variety of disorders [7,8,9]. Two of the fifteen known human (h) CA isoforms, hCA II and VII, are key cytosolic isoforms involved in brain metabolism and neuronal excitation [10]. Consequently, isoform-selective hCA II/VII inhibitors are identified as potential therapeutic targets for neurological diseases and disorders such as epilepsy, seizures, and Alzheimer disease [11,12]. In this context, inhibition of these isozymes was recently proposed as a new approach for the management of neuropathic pain [13,14,15]. It should be noted that the lack of approved medicines for the treatment of neuropathic pain as well as many other conditions in which CA activity is unbalanced is one of the major challenges in medicine [16,17,18,19,20,21,22,23,24,25,26,27,28,29,30,31,32,33,34,35,36,37,38,39,40]. Due to their unique zinc-binding properties as anions, primary sulfonamides (-SO_2_NH_2_) are the main classes of CAs inhibitors (CAIs) [16,17,18,19,20,21,22,23,24,25,26] and, not surprisingly, the majority of reported CA inhibitors (CAIs) contain at least one sulfonamide moiety in their structures [27,28,29]. Very recently, our group disclosed that the clinically used antibiotic Furagin (Figure 1a), which contains hydantoin moiety, shows effective inhibitory activity on several hCAs [30]. Along this line, we herein extend this earlier investigation to series of 1-(4-benzenesulfonamide)-3-alkyl/benzyl-hydantoin derivatives, with special emphasize on their inhibitory effects against CA II and VII (Figure 1b). The newly developed compounds were also tested for the inhibition of two β-CAs from fungal pathogens. Indeed, in many pathogenic bacteria [41,42,43,44,45,46,47] and fungi [48,49,50,51,52], CAs belonging to several genetic familieshaverelevant physiologic functions and their inhibition may lead to anti-infective effects [53,54,55,56,57].

## 2. Results and Discussion

### 2.1. Compounds Design and Synthesis

Considering the fact that hydantoins already possess CA inhibitory effects [30], the drug design strategy that we propose in this paper is to incorporate in the same molecule both a zinc binder fragment of the benzene-sulfonamide type [4,5,6,7,8,9,16,17,18] as well as the tail based on the 3-substituted-hydantoin motif.

The synthesis of the target 1-(4-benzenesulfonamide)-3-alkyl/benzyl-hydantoin derivatives is shown in Figure 1. The synthesis started from sulfanilamide **1,** which was converted to 4-thioureidobenzenesulfonamide (**2**)via reaction with KSCN in aqueous, acidic medium [33]. The key intermediate, potassium cyano(4-sulfamoylphenyl)amide **4,** was prepared by the selective *S*-methylation of thiourea **2** via treatment with 1 equiv. of MeI, followed by elimination of metheylthiolate from the formed methyl (4-sulfamoylphenyl)carbamimidothioate (**3**) by treatment with K_2_CO_3_ at elevated temperature. Subsequently, intermediate **4** was treated with ethyl 2-bromoacetate, leading to **5,** which was treated with hydrochloric acid at an elevated temperature, thus affording 4-(2,4-dioxoimidazolidin-1-yl)benzenesulfonamide (**6**). In the final step, the selective *N*-alkylation/benzylation of the NH hydantoin moiety with various alkyl/allyl/benzyl-halides (**7a**–**n**) provided the desired compounds (**8a**–**n**) in acceptable to good yield. ^1^H NMR, ^13^C NMR, and HRMS techniques were used to confirm the chemical structure of all of the synthesized compounds. All the analyzed compounds were >95% HPLC pure.

### 2.2. Carbonic Anhydrase Inhibition

The new compounds designed here were tested as inhibitors of three human enzymes, i.e., isoforms hCA I, II, and VII (all cytosolic ones) [4,5,6,7,8,9,16,17,18], as well as two fungal β-CAs from pathogenic organisms: MgCA from *Malassezia globosa*, one of the fungi involved in dandruff formation [58,59,60,61]; and CgNce103 from *Candida glabrata*, a species known for its virulence and resistance to many classes of antifungal drugs in clinical use [62,63,64,65,66]. The classical sulfonamide CAI acetazolamide (5-acetamido-1,3,4-thiadiazole-2-sulfonamide, **AAZ**) was used as standard in the measurements reported in Table 1.

Data of Table 1 show the following structure-activity relationship (SAR) for the inhibition of these enzymes with hydantoin-substituted benzene-sulfonamides:(i)hCA I, an abundant cytosolic isoform in many tissues and organs [4,5,6,7,8,9], was moderately inhibited by compounds **6** and **8** investigated here, with K_I_ ranging between 233.8 and 8789 nM. Some of the best hCA I inhibitors are as active as **AAZ**, the standard drug (Table 1).(ii)hCA II; the dominant cytosolic isoform [4,5,6,7,8,9] was, on the other hand, potently inhibited by most new sulfonamides reported here, with K_I_ ranging between 1.2 and 91.2 nM. The best inhibitor **8l** incorporates the 2-fluorobenzyl moiety in position 3 of the hydantoin ring, whereas the unsubstituted benzyl derivative **8d** was also a highly effective inhibitor (K_I_ of 8.7 nM). The alkyl or alkenyl substituted derivatives **8a**–**8c** were slightly less effective (but still potent CAIs), whereas the position and nature of the substituent eventually present on the benzyl fragment in the remaining derivatives seemed to be the factor that strongly influenced the inhibition potency. Indeed, 4-CN, 4-nitro and 2-fluorobenzyl fragments were those associated with the best inhibitory action, whereas 3-methyl, pentafluoro, 4-CF_3_ and 4-Cl led to less effective inhibitors.(iii)The SAR is rather different for the inhibition of CA VII. The unsubstituted hydantoin **6** and the alky-substituted ones, **8a** and **8b**, were moderately active (K_I_ of 30.8–187.2 nM). The alkeyl and benzylsubstituted hydantoins (except **8m**) were, on the other hand, effective hCA VII inhibitors, with K_I_ ranging between 3.0–19.5 nM. The best hCA VII inhibitors were the unsubstituted benzyl and the 4-Me-benzyl derivatives **8d** and **8e**, with K_I_ of 3.0–5.3 nM, in the same range as AAZ.(iv)MgCA was poorly inhibited by these sulfonamides, which had some activity in the high micromolar range, similarly to AAZ (Table 1).(v)CgNce103 was, on the other hand, effectively inhibited by hydantoin-substituted benzene-sulfonamides, with K_I_ ranging between 5.9 and 83.7 nM. The SAR is again diverse from what observed for other isoforms/enzymes. The unsubstituted hydantoin **6** and the alkyl-substituted derivatives **8a**–**8c** showed K_I_ of 29.5–83.7 nM, whereas most benzyl-substituted derivatives (except **8l** and **8m**) were active in the low nanomolar range.

## 3. Materials and Methods

### 3.1. Chemistry

Reagents, starting materials and solvents were obtained from commercial sources and used as received. Thin-layer chromatography was performed on silica gel, spots were visualized with UV light (254 and 365 nm). NMR spectra were recorded on Bruker 300 spectrometer with chemical shifts values (δ) in ppm relative to TMS using the residual DMSO-d_6_ signal (^1^H 2.50; ^13^C 39.52) see also Appendix A. High-resolution mass spectra (HRMS) were recorded on a mass spectrometer with a Q-TOF micro mass analyzer using the ESI technique.

### 3.2. Synthesis

#### 3.2.1. 4-Thioureidobenzenesulfonamide (**2**)

4-Aminobenzensulfonamide (**1**) (30 g, 174.3 mmol) was dissolved in aqueous HCl (3.5 M, 180 mL) at 70 °C. After cooling to room temperature, KSCN (16.94 g, 174.3 mmol) was added, and the mixture was refluxed for 3 h. After cooling to room temperature, the reaction mixture was poured onto ice/cold water, and the formed precipitate was collected by filtration, washed with water, and air dried to afford **2** (12.49 g, 31%) as a white powder.

^1^H NMR (300 MHz, DMSO-d_6_) *δ* = 7.32 (s, 2H), 7.69 (d, 2H, *J =* 8.6 Hz), 7.77 (d, 2H, *J =* 8.6 Hz), 10.02 (s, 1H) ppm ^13^C NMR (75 MHz, DMSO-d_6_) *δ* = 122.8, 127.3, 139.8, 143.9, 182.8 ppm MS (ESI) [M + H]^+^: *m*/*z* 232.0.



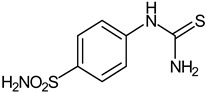



#### 3.2.2. Methyl (4-Sulfamoylphenyl)carbamimidothioate (**3**)

To a solution of 4-thioureidobenzenesulfonamide (**2**) (300 mg, 1.3 mmol) in DMF (4 mL),MeI (0.08 mL, 1.3 mmol) was added, and the mixture was heated at 40 °C for 2.5 h. After cooling to room temperature, the reaction mixture was extracted with EtOAc (3 × 20 mL). Organic layer was washed with aq. sat. NaHCO_3_ (2 × 20 mL) and then aq. sat. NH_4_Cl (1 × 20 mL), and dried over Na_2_SO_4_. Solvent removal in vacuum resulted in **3** (223 mg, 70%) as a white powder.

^1^H NMR (300 MHz, DMSO-d_6_) *δ* = 2.37 (s, 3H), 6.63 (s, 2H), 6.94 (s, 2H), 7.22 (s, 2H), 7.71 (d, 2H, *J =* 8.4 Hz) ppm ^13^C NMR (75 MHz, DMSO-d_6_) *δ* = 14.2, 122.8, 127.7, 138.0, 153.9, 157.0 ppm HRMS (ESI) [M + H]^+^: *m*/*z* calcd for (C_8_H_12_N_3_O_2_S_2_) 246.0371. Found 246.0372.



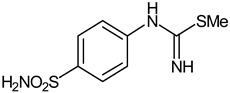



#### 3.2.3. Potassium Cyano(4-sulfamoylphenyl)amide (**4**)

To a solution of methyl (4-sulfamoylphenyl) carbamimidothioate (**3**) (500 mg, 2.04 mmol) in DMF (8 mL), K_2_CO_3_ (564 mg, 4.08 mmol) was added, and the mixture was stirred at 100 °C for 1.5 h. The mixture was cooled to room temperature and precipitate was removed by filtration. To the filtrate, EtOAc (80 mL) was added and precipitate formed was collected by filtration, washed with EtOAc (20 mL), and air dried to afford **4** (427 mg, 89%) as a white powder.

^1^H NMR (300 MHz, DMSO-d_6_) *δ* = 6.60 (d, 2H, *J =* 8.6 Hz), 6.85 (s, 2H), 7.29 (s, 1H), 7.38 (d, 2H, *J* = 8.6 Hz) ppm ^13^C NMR (75 MHz, DMSO-d_6_) *δ* = 118.0, 125.7, 127.9, 129.0, 160.9 ppm HRMS (ESI) [M − K]^−^: *m*/*z* calcd for (C_7_H_6_N_3_O_2_S) 196.0181. Found 196.0188.



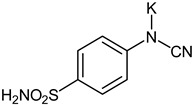



#### 3.2.4. 4-(2,4-Dioxoimidazolidin-1-yl)benzenesulfonamide (**6**)

To a suspension of potassium cyano(4-sulfamoylphenyl)amide(**4**)(4.0 g, 17 mmol) in MeOH (90 mL), ethyl 2-bromoacetate (1.76 mL, 17 mmol) was added dropwise. The mixture was heated at 65 °C for 3.5 h. After cooling to room temperature conc. HCl (11.25 mL) was dropwise added, and the mixture was stirred for 2.5 h at 65 °C. The solvent was evaporated under reduced pressure and the residue was washed with *i*PrOH (50 mL) and dried in vacuum to afford **6** (3.98 g, 92%) as a white powder.

^1^H NMR (300 MHz, DMSO-d_6_) *δ* = 4.51 (s, 2H), 7.34 (s, 2H), 7.78–7.85 (m, 4H), 11.40 (s, 1H) ppm ^13^C NMR (75 MHz, DMSO-d_6_) *δ* = 51.9, 118.4, 127.6, 139.2, 141.9, 155.9, 171.1 ppm HRMS (ESI) [M − 1]^−^: *m*/*z* calcd for (C_9_H_8_N_3_O_4_S) 254.0236. Found 254.0239.



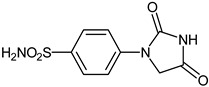



#### 3.2.5. 4-(3-Ethyl-2,4-dioxoimidazolidin-1-yl)benzenesulfonamide (**8a**)

To a stirred solution of 4-(2,4-dioxoimidazolidin-1-yl)benzenesulfonamide (**6**) (250 mg, 0.98 mmol) and ethyl iodide (0.079 mL, 0.98 mmol) in DMF (5 mL) K_2_CO_3_ (270 mg, 1.96 mmol) was added at room temperature and the mixture was stirred at this temperature for 5 h. It was extracted with DCM (3 × 20 mL), the organic phase was dried over Na_2_SO_4_, and volatiles were removed in vacuum to afford **8a** (103 mg, 37%) as a white solid.

^1^H NMR (300 MHz, DMSO-d_6_) *δ* = 1.16 (t, 3H, *J =* 7.1 Hz), 3.51 (q, 2H, *J =* 7.1), 4.50 (s, 2H), 7.35 (s, 2H), 7.78–7.86 (m, 4H) ppm ^13^C NMR (125 MHz, DMSO-d_6_) *δ* = 14.3, 34.7, 51.0, 119.0, 128.2, 139.5, 142.1, 155.6, 170.1 ppm HRMS (ESI) [M − 1]^−^: *m*/*z* calcd for (C_11_H_12_N_3_O_4_S) 282.0549. Found 282.0557.



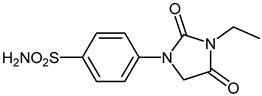



#### 3.2.6. 4-(3-Heptyl-2,4-dioxoimidazolidin-1-yl)benzenesulfonamide (**8b**)

To a stirred solution of 4-(2,4-dioxoimidazolidin-1-yl)benzenesulfonamide (**6**) (250 mg, 0.98 mmol) and 1-iodoheptane (0.160 mL, 0.98 mmol) in DMF (5 mL), K_2_CO_3_ (270 mg, 1.96 mmol) was added at room temperature and the mixture was stirred at this temperature for 5 h. Water was added to the reaction mixture and the precipitate former was collected by filtration, washed with water, and air dried to afford **8b** (109 mg, 32%) as a white solid.

^1^H NMR (300 MHz, DMSO-d_6_) *δ* = 0.90 (t, 3H, *J =* 6.6 Hz), 1.31 (br. s, 8H), 1.53–1.62 (m, 2H), 3.47 (t, 2H, *J =* 6.6 Hz), 4.53 (s, 2H), 7.35 (s, 2H), 7.78–7.86 (m, 4H) ppm ^13^C NMR (75 MHz, DMSO-d_6_) *δ* = 14.8, 22.9, 27.0, 28.3, 29.1, 32.0, 39.1, 50.6, 118.4, 127.7, 139.4, 141.7, 155.3, 169.7 ppm HRMS (ESI) [M − 1]^−^: *m*/*z* calcd for (C_16_H_22_N_3_O_4_S) 352.1331. Found 352.1341.



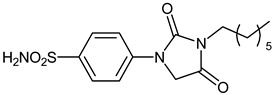



#### 3.2.7. 4-(3-Allyl-2,4-dioxoimidazolidin-1-yl)benzenesulfonamide (**8c**)

To a stirred solution of 4-(2,4-dioxoimidazolidin-1-yl)benzenesulfonamide (**6**) (250 mg, 0.98 mmol) and allyl bromide (0.085 mL, 0.98 mmol) in DMF (5 mL), K_2_CO_3_ (270 mg, 1.96 mmol) was added at room temperature, and the mixture was stirred at this temperature for 3 h. Water was added to the reaction mixture and it was extracted with DCM (3 × 20 mL), the organic phase was dried over Na_2_SO_4_, and the solvent was evaporated in vacuum to give **8c** (156 mg, 54%) as a white solid.

^1^H NMR (300 MHz, DMSO-d_6_) *δ* = 4.12 (d, 2H, *J* = 3.4 Hz), 4.61 (s, 2H), 5.17–5.25 (m, 2H), 5.82–5.92 (m, 1H), 7.35 (s, 2H), 7.82–7.89 (m, 4H) ppm ^13^C NMR (75 MHz, DMSO-d_6_) *δ* = 50.7, 117.7, 118.5, 127.7, 132.7, 139.5, 141.7, 154.9, 169.4 ppm HRMS (ESI) [M − 1]^−^: *m*/*z* calcd for (C_12_H_12_N_3_O_4_S) 294.0549. Found 294.0551.



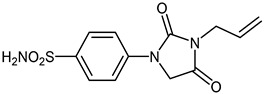



#### 3.2.8. 4-(3-Benzyl-2,4-dioxoimidazolidin-1-yl)benzenesulfonamide (**8d**)

To a stirred solution of 4-(2,4-dioxoimidazolidin-1-yl)benzenesulfonamide (**6**) (250 mg, 0.98 mmol) and benzyl bromide (0.116 mL, 0.98 mmol) in DMF (5 mL), K_2_CO_3_ (270 mg, 1.96 mmol) was added at room temperature, and the mixture was stirred at this temperature for 5 h. Water was added to the reaction mixture and the precipitate formed was collected by filtration, washed with water and Et_2_O, and air dried to afford **8d** (205 mg, 61%) as a white solid.

^1^H NMR (300 MHz, DMSO-d_6_) *δ* = 4.66 (s, 2H), 4.70 (s, 2H), 7.32–7.40 (m, 7H), 7.81–7.90 (m, 4H) ppm ^13^C NMR (75 MHz, DMSO-d_6_) *δ* = 42.6, 50.9, 118.6, 127.7, 128.5, 128.5, 129.4, 137.0, 139.5, 141.6, 155.1, 169.7 ppm HRMS (ESI) [M − 1]^−^: *m*/*z* calcd for (C_16_H_14_N_3_O_4_S) 344.0705. Found 344.0708.



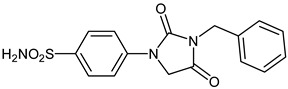



#### 3.2.9. 4-(3-(4-Methylbenzyl)-2,4-dioxoimidazolidin-1-yl)benzenesulfonamide (**8e**)

To a stirred solution of 4-(2,4-dioxoimidazolidin-1-yl)benzenesulfonamide (**6**) (250 mg, 0.98 mmol) and 4-methylbenzyl bromide (181 mg, 0.98 mmol) in DMF (5 mL) K_2_CO_3_ (270 mg, 1.96 mmol) was added at room temperature and the mixture was stirred at this temperature for 5 h. Water was added to the reaction mixture and precipitate formed was collected by filtration, washed with water and Et_2_O and air dried to afford **8e** (179 mg, 51%) as a white solid.

^1^H NMR (300 MHz, DMSO-d_6_) *δ* = 2.31 (s, 3H), 4.55–4.64 (m, 4H), 7.18 (d, 2H, *J* = 12.3 Hz), 7.28 (d, 2H, *J* = 12.3 Hz), 7.36 (s, 2H), 7.76–7.84 (m, 4H) ppm ^13^C NMR (75 MHz, DMSO-d_6_) *δ* = 21.6, 42.4, 50.9, 118.6, 127.8, 128.6, 130.0, 134.1, 137.7, 139.6, 141.6, 155.1, 169.6 ppm HRMS (ESI) [M − 1]^−^: *m*/*z* calcd for (C_17_H_16_N_3_O_4_S) 358.0862. Found 358.0869.



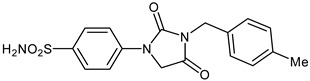



#### 3.2.10. 4-(3-(4-Chlorobenzyl)-2,4-dioxoimidazolidin-1-yl)benzenesulfonamide (**8f**)

To a stirred solution of 4-(2,4-dioxoimidazolidin-1-yl)benzenesulfonamide (**6**) (250 mg, 0.98 mmol) and 4-chlorobenzyl bromide (201 mg, 0.98 mmol) in DMF (5 mL), K_2_CO_3_ (270 mg, 1.96 mmol) was added at room temperature, and the mixture was stirred at this temperature for 3.5 h. Water was added to the reaction mixture and precipitate formed was collected by filtration, washed with water and DCM, and air dried to afford **8f** (137 mg, 37%) as a white solid.

^1^H NMR (300 MHz, DMSO-d_6_) *δ* = 4.64 (s, 2H), 4.69 (s, 2H), 7.37 (s, 2H), 7.40–7.47 (m, 4H), 7.82–7.89 (m, 4H) ppm ^13^C NMR (75 MHz, DMSO-d_6_) *δ* = 41.9, 50.9, 118.5, 127.7, 129.4, 130.5, 133.1, 136.0, 139.6, 141.6, 155.0, 169.6 ppm HRMS (ESI) [M–1]^−^: *m*/*z* calcd for (C_16_H_13_N_3_O_4_SCl) 378.0315. Found 378.0320.



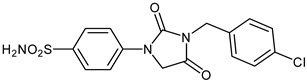



#### 3.2.11. 4-(3-(4-Cyanobenzyl)-2,4-dioxoimidazolidin-1-yl)benzenesulfonamide (**8g**)

To a stirred solution of 4-(2,4-dioxoimidazolidin-1-yl)benzenesulfonamide (**6**) (250 mg, 0.98 mmol) and 4-cyanobenzyl bromide (192 mg, 0.98 mmol) in DMF (5 mL), K_2_CO_3_ (270 mg, 1.96 mmol) was added at room temperature, and the mixture was stirred at this temperature for 3 h. Water was added to the reaction mixture and precipitate formed was collected by filtration, washed with water and Et_2_O, and air dried to afford **8g** (184 mg, 51%) as a white solid.

^1^H NMR (300 MHz, DMSO-d_6_) *δ* = 4.66 (s, 2H), 4.80 (s, 2H), 7.37 (s, 2H), 7.60 (d, 2H, *J =* 7.9 Hz), 7.82–7.91 (m, 6H) ppm ^13^C NMR (75 MHz, DMSO-d_6_) *δ* = 42.3, 51.0, 111.2, 118.5, 119.6, 127.7, 129.2, 133.4, 139.6, 141.6, 142.6, 155.0, 169.7 ppm HRMS (ESI) [M − 1]^−^: *m*/*z* calcd for (C_17_H_13_N_4_O_4_S) 369.0658. Found 369.0663.



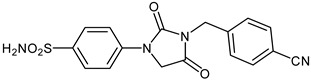



#### 3.2.12. 4-(3-(4-Nitrobenzyl)-2,4-dioxoimidazolidin-1-yl)benzenesulfonamide (**8h**)

To a stirred solution of 4-(2,4-dioxoimidazolidin-1-yl)benzenesulfonamide (**6**) (250 mg, 0.98 mmol) and 4-nitrobenzyl bromide (211 mg, 0.98 mmol) in DMF (5 mL), K_2_CO_3_ (270 mg, 1.96 mmol) was added at room temperature, and the mixture was stirred at this temperature for 5 h. Water was added to the reaction mixture and precipitate formed was collected by filtration, washed with water and Et_2_O, and air dried to afford **8h** (188 mg, 49%) as a white solid.

^1^H NMR (300 MHz, DMSO-d_6_) *δ* = 4.67 (s, 2H), 4.85 (s, 2H), 7.37 (s, 2H), 7.68 (d, 2H, *J* = 7.2 Hz), 7.83–7.91 (m, 4H), 8.25 (d, 2H, *J =* 7.2 Hz) ppm ^13^C NMR (75 MHz, DMSO-d_6_) *δ* = 42.1, 51.0, 118.5, 124.5, 127.7, 129.6, 139.6, 141.6, 144.7, 147.8, 155.0, 169.7 ppm HRMS (ESI) [M − 1]^−^: *m*/*z* calcd for (C_16_H_13_N_4_O_6_S) 389.0556. Found 389.0556.



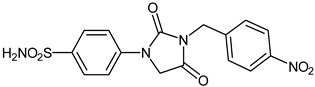



#### 3.2.13. 4-(2,4-Dioxo-3-(4-(trifluoromethyl)benzyl)imidazolidin-1-yl)benzenesulfonamide (**8i**)

To a stirred solution of 4-(2,4-dioxoimidazolidin-1-yl)benzenesulfonamide (**6**) (250 mg, 0.98 mmol) and 4-(trifluoromethyl)benzyl bromide (234 mg, 0.98 mmol) in DMF (5 mL), K_2_CO_3_ (270 mg, 1.96 mmol) was added at room temperature and the mixture was stirred at this temperature for 3 h. Water was added to the reaction mixture and precipitate formed was collected by filtration, washed with water and Et_2_O, and air dried to afford **8i** (138 mg, 34%) as a white solid.

^1^H NMR (500 MHz, DMSO-d_6_) *δ* = 4.66 (s, 2H), 4.80 (s, 2H, 7.36 (s, 2H), 7.63 (d, 2H, *J =* 7.2 Hz), 7.76 (d, 2H, *J =* 7.2 Hz), 7.84–7.89 (m, 4H) ppm ^13^C NMR (125 MHz, DMSO-d_6_) *δ* = 42.2, 51.0, 118.5, 125.1 (q, *J* = 271.9 Hz) 126.3, 126.4, 127.7, 129.1 (q, *J* = 31.4 Hz) 129.3, 139.6, 141.6, 141.7, 155.0, 169.7 ppm ^19^F NMR (470 MHz) *δ* = –60.9 ppm HRMS (ESI) [M − 1]^−^: *m*/*z* calcd for (C_17_H_13_N_3_O_4_F_3_S) 412.0579. Found 412.0597.



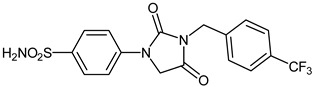



#### 3.2.14. 4-(2,4-Dioxo-3-(4-(trifluoromethoxy)benzyl)imidazolidin-1-yl)benzenesulfonamide (**8j**)

To a stirred solution of 4-(2,4-dioxoimidazolidin-1-yl)benzenesulfonamide (**6**) (250 mg, 0.98 mmol) and 4-(trifluoromethoxy)benzyl bromide (0.157 mL, 0.98 mmol) in DMF (5 mL), K_2_CO_3_ (270 mg, 1.96 mmol) was added at room temperature, and the mixture was stirred at this temperature for 3 h. Water was added to the reaction mixture and precipitate formed was collected by filtration, washed with water and Et_2_O, and air dried to afford **8j** (226 mg, 54%) as a white solid.

^1^H NMR (500 MHz, DMSO-d_6_) *δ* = 4.65 (s, 2H), 4.73 (s, 2H), 7.36 (s, 2H), 7.39 (d, 2H, *J =* 7.2 Hz), 7.54 (d, 2H, *J =* 7.2 Hz), 7.83–7.89 (4H, m) ppm ^13^C NMR (125 MHz, DMSO-d_6_) *δ* = 41.9, 50.9, 118.5, 122.0, 121.0 (q, *J* = 256.0 Hz), 127.7, 130.6, 136.5, 139.6, 141.6, 148.6, 155.0, 169.6 ppm ^19^F NMR (470 MHz) *δ* = –56.8 ppm HRMS (ESI) [M − 1]^−^: *m*/*z* calcd for (C_17_H_13_N_3_O_5_SF_3_) 428.0528. Found 428.0533.



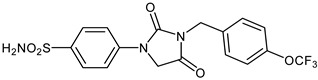



#### 3.2.15. 4-(3-(3-Methylbenzyl)-2,4-dioxoimidazolidin-1-yl)benzenesulfonamide (**8k**)

To a stirred solution of 4-(2,4-dioxoimidazolidin-1-yl)benzenesulfonamide (**6**) (250 mg, 0.98 mmol) and 3-methylbenzyl bromide (0.133 mL, 0.98 mmol) in DMF (5 mL), K_2_CO_3_ (270 mg, 1.96 mmol) was added at room temperature, and the mixture was stirred at this temperature for 5 h. Water was added to the reaction mixture and precipitate formed was collected by filtration, washed with water and Et_2_O, and air dried to afford **8k** (120 mg, 34%) as a white solid.

^1^H NMR (300 MHz, DMSO-d_6_) *δ* = 2.32 (s, 3H), 4.65 (s, 2H), 4.66 (s, 2H), 7.15–7.29 (m, 4H), 7.36 (s, 2H), 7.80–7.89 (m, 4H) ppm ^13^C NMR (75 MHz, DMSO-d_6_) *δ* = 21.4, 42.0, 50.3, 118.0, 125.2, 127.2, 128.6, 128.8, 136.5, 138.1, 139.0, 141.1, 154.6, 169.1 ppm HRMS (ESI) [M − 1]^−^: *m*/*z* calcd for (C_17_H_16_N_3_O_4_S) 358.0862. Found 358.0869.



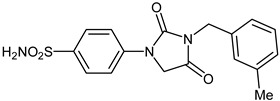



#### 3.2.16. 4-(3-(2-Fluorobenzyl)-2,4-dioxoimidazolidin-1-yl)benzenesulfonamide (**8l**)

To a stirred solution of 4-(2,4-dioxoimidazolidin-1-yl)benzenesulfonamide (**6**) (250 mg, 0.98 mmol) and 2-fluorobenzyl bromide (0.118 mL, 0.98 mmol) in DMF (5 mL), K_2_CO_3_ (270 mg, 1.96 mmol) was added at room temperature, and the mixture was stirred at this temperature for 5 h. Water was added to the reaction mixture and precipitate formed was collected by filtration, washed with water and Et_2_O, and air dried to afford **8l** (164 mg, 46%) as a white solid.

^1^H NMR (500 MHz, DMSO-d_6_) *δ* = 4.66 (s, 2H), 4.75 (s, 2H), 7.20–7.27 (m, 2H), 7.36 (s, 2H), 7.37–7.47 (m, 2H), 7.83–7.88 (m, 4H) ppm ^13^C NMR (125 MHz, DMSO-d_6_) *δ* = 36.5 (d, *J =* 4.6 Hz), 50.9, 116.2 (d, *J =* 20.9 Hz), 118.5, 123.6 (d, *J* = 14.2 Hz), 125.3 (d, *J* = 3.4 Hz), 127.7, 130.6 (d, *J =* 8.1 Hz), 130.7 (d, *J =* 3.6 Hz), 139.6, 141.6, 154.9, 160.8 (d, *J =* 245.9 Hz), 169.5 ppm ^19^F NMR (470 MHz) –118.0 ppm HRMS (ESI) [M − 1]^−^: *m*/*z* calcd for (C_16_H_13_N_3_O_4_FS) 362.0611. Found 362.0619.



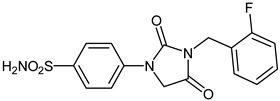



#### 3.2.17. 4-(3-(3,4-Dichlorobenzyl)-2,4-dioxoimidazolidin-1-yl)benzenesulfonamide (**8m**)

To a stirred solution of 4-(2,4-dioxoimidazolidin-1-yl)benzenesulfonamide (**6**) (250 mg, 0.98 mmol) and 3,4-dichlorobenzyl bromide (0.142 mL, 0.98 mmol) in DMF (5 mL), K_2_CO_3_ (270 mg, 1.96 mmol) was added at room temperature, and the mixture was stirred at this temperature for 2.5 h. Water was added to the reaction mixture and precipitate formed was collected by filtration, washed with water and Et_2_O, and air dried to afford **8m** (194 mg, 48%) as a white solid.

^1^H NMR (500 MHz, DMSO-d_6_) *δ* = 4.64 (s, 2H), 4.71 (s, 2H), 7.36 (s, 2H), 7.40 (d, 1H, *J =* 8.6 Hz), 7.66 (d, 2H, *J =* 8.6 Hz), 7.81–7.90 (m, 4H) ppm ^13^C NMR (75 MHz, DMSO-d_6_) *δ* = 41.5, 51.0, 118.5, 127.7, 128.9, 130.5, 131.1, 131.5, 132.0, 138.1, 139.6, 141.6, 155.0, 169.7 ppm HRMS (ESI) [M − 1]^−^: *m*/*z* calcd for (C_16_H_12_N_3_O_4_SCl_2_) 411.9926. Found 411.9933.



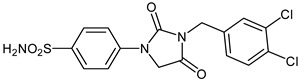



#### 3.2.18. 4-(2,4-Dioxo-3-((perfluorophenyl)methyl)imidazolidin-1-yl)benzenesulfonamide (**8n**)

To a stirred solution of 4-(2,4-dioxoimidazolidin-1-yl)benzenesulfonamide (**6**) (250 mg, 0.98 mmol) and 2,3,4,5,6-pentafluorobenzyl bromide (0.148 mL, 0.98 mmol) in DMF (5 mL), K_2_CO_3_ (270 mg, 1.96 mmol) was added at room temperature and the mixture was stirred at this temperature for 5 h. Water was added to the reaction mixture and precipitate formed was collected by filtration, washed with water and Et_2_O, and air dried to afford **8n** (229 mg, 54%) as a white solid.

^1^H NMR (500 MHz, DMSO-d_6_) *δ* = 4.59 (s, 2H), 4.81 (s, 2H), 7.35 (s, 2H), 7.81–7.87 (m, 4H) ppm ^13^C NMR (125 MHz, DMSO-d_6_) *δ* = 36.5 (d, *J =* 4.6 Hz), 50.9, 116.2 (d, *J =* 20.9 Hz), 118.5, 123.6 (d, *J =* 14.2 Hz), 125.3 (d, *J =* 3.4 Hz), 127.7, 130.6 (d, *J =* 8.1 Hz), 130.7 (d, *J* = 3.6 Hz), 139.6, 141.6, 154.9, 160.8 (d, *J* = 245.9 Hz), 169.5 ppm ^19^F NMR (470 MHz) *δ* = −140.8 (dd, 2F, *J* = 16.5, 6.3 Hz), −155.1 (t, 1F, *J =* 21.9 Hz), −163.2–−163.3 (2F, m) ppm HRMS (ESI) [M − 1]^−^: *m*/*z* calcd for (C_16_H_9_N_3_O_4_F_5_S) 434.0234. Found 434.0246.



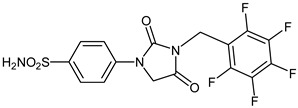



### 3.3. CA Inhibition Assay

An applied photophysics stopped-flow instrument was used for assaying the CA catalysed CO_2_ hydration activity [67]. Phenol red (at a concentration of 0.2 mM) was used as indicator, working at the absorbance maximum of 557 nm, with 20 mM Hepes (pH 7.5) as buffer for α-CAs or 20 mM TRIS (pH 8.4) as buffer for β-CAs, and 20 mM Na_2_SO_4_ (for maintaining constant the ionic strength), following the initial rates of the CA-catalysed CO_2_ hydration reaction for a period of 10–100 s. The CO_2_ concentrations ranged from 1.7 to 17 mM for the determination of the kinetic parameters and inhibition constants. For each inhibitor, at least six traces of the initial 5–10% of the reaction were used for determining the initial velocity. The uncatalysed rates were determined in the same manner and subtracted from the total observed rates. Stock solutions of inhibitor (0.1 mM) were prepared in distilled–deionised water, and dilutions up to 0.01 nM were done thereafter with the assay buffer. Inhibitor and enzyme solutions were preincubated together for 6 h at room temperature prior to assay in order to allow for the formation of the E–I complex. The inhibition constants were obtained by nonlinear least-squares methods using PRISM 3 and the Cheng–Prusoff equation, as reported earlier [68,69,70,71,72,73,74], and represent the mean from at least three different determinations. All CA isoforms were recombinant ones obtained in-house as reported earlier [25,58,59,60,61,66,75], and their concentrations in the assay system ranged between 9–12 nM.

## 4. Conclusions

Starting from commercially available inexpensive 4-aminobenzenesulfonamide, a library of novel hydantoin-based benzenesulfonamides were synthesized, and the structures of all derivatives were confirmed by ^1^H NMR, ^13^C NMR, and HRMS spectral techniques. The prepared compounds were screened for their hCA inhibitory activities against three cytosolic isoforms as well as two β-CAs from fungal pathogens. The best inhibition was observed against hCA II and VII, as well as *Candida glabrata* enzyme CgNce103. hCA I and MgCA were, on the other hand, less effectively inhibited by these compounds. The interesting results against CgNce103 make the compounds of interest for investigations in vivo as potential antifungals.

## Data Availability

Data are contained within the article.

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
