# Peer review of "Benzenesulfonamides Incorporating Hydantoin Moieties Effectively Inhibit Eukaryoticand Human Carbonic Anhydrases"

_ijms, 2022, doi:10.3390/ijms232214115_

Round 1

Reviewer 1 Report

The topic of manuscript is relevant, and  it is quite good written, however some parts of the work should be improved: 

1) The name of the manuscript should be changed. Authors didn't tested inhibitors against prokaryotic carbonic anhydrases (CA). They worked only with human and Yeast CA and yeast are eukaryotes. Line 53 should be corrected too.

2) line 365: “All CA isoforms were recombinant ones obtained in-house as reported earlier [19,21,25,26]”. Authors should supplement information about preparation of enzymes, because 25-26 references is about bacterial enzymes (while authors doesn’t used any of bacterial CA), reference 19 is about fungal CAs, but it does not include description of recombinant CA production, and reference 20 describes the production of only recombinant CgNCE103. So what about Human CAs and CA from Malassezia?

Also I would like to ask:

1) line 353: “Phenol red (at a concentration of 0.2 mM) was used as indicator, working at the absorbance maximum of 557 nm, with 20 mM Hepes (pH 7.5) as buffer for a-CAs or 20 mM TRIS (pH 8.4) as buffer for ß-CAs, <…>” Why different pH buffers, were used for Alfa and Beta CAs?

Author Response

  • In accordance with your suggestion, the title and related text (Line 53) were revised.
  • References 25 and 26 were deleted. A new reference for the production of M. Globosa was added [75]. For the production of Human CAs, reference 25 was highlighted.
  • At pH values < 8.3 the active sites of beta carbonic anhydrases are closed and they are in their inactive form (are not catalytically active). However, at pH values > 8.3, the “closed active site” is converted to the “open active site” (with gain of catalytic activity).

Reviewer 2 Report

The manuscript presents a logical study of compounds against carbonic anhydrases, an important target for drug development against various diseases. I would like to see the supplementary information containing NMR and other data for the synthesized compounds before making a conclusion whether this work can be published in this journal or not.

Following are the comments which can be addressed to make manuscript better.

Comments:

1.      In the title of article “prokaryoric” should be replaced with prokaryotic

2.      Change the sentence starting from “In this context….is unbalanced.” for more clarity e.g., split it into two sentences.

3.      In scheme 1 remove the double numbering 4 and yield percentage, mention the yields in the scheme legends and describe whether the yield of step V is after two steps.

4.      There is difference in the notation of inhibition data in the legends of table KI is mentioned whereas in the text both KIS and KI is mentioned, use uniform notation for better understanding.

Author Response

  • The title was revised.
  • The mentioned sentence was splite into two sentences.
  • Scheme 1 was revised.
  • In accordance with your suggestion, KIS was replaced with KI.
